# Mucin4 (MUC4) Antibody Labeled with an NIR Dye Brightly Targets Pancreatic Cancer Liver Metastases and Peritoneal Carcinomatosis

**DOI:** 10.3390/cancers17122031

**Published:** 2025-06-18

**Authors:** Sunidhi Jaiswal, Siamak Amirfakhri, Javier Bravo, Keita Kobayashi, Abhijit Aithal, Sumbal Talib, Kavita Mallya, Maneesh Jain, Aaron M. Mohs, Robert M. Hoffman, Surinder K. Batra, Michael Bouvet

**Affiliations:** 1Department of Surgery, University of California San Diego, La Jolla, CA 92093, USA; s3jaiswal@health.ucsd.edu (S.J.); siamirfakhri@ucsd.edu (S.A.); jbquintana@ucsd.edu (J.B.); kjkobayashi@ucsd.edu (K.K.); rhoffman@health.ucsd.edu (R.M.H.); 2VA San Diego Healthcare System, La Jolla, CA 92093, USA; 3Department of Biochemistry and Molecular Biology, University of Nebraska Medical Center, Omaha, NE 68105, USA; abhijit.aithal@unmc.edu (A.A.); kmallya@unmc.edu (K.M.); mjain@unmc.edu (M.J.); aaron.mohs@unmc.edu (A.M.M.); sbatra@unmc.edu (S.K.B.); 4Department of Pharmaceutical Sciences, University of Nebraska Medical Center, Omaha, NE 68105, USA; stalib@unmc.edu; 5AntiCancer Inc., San Diego, CA 92111, USA

**Keywords:** pancreatic cancer, mouse model, orthotopic, liver metastasis, tumor targeting, MUC4, antibody, fluorescence, peritoneal carcinomatosis, NIR Dye, mucins

## Abstract

Pancreatic ductal adenocarcinoma (PDAC) is a highly lethal disease. The most effective cure is surgical resection, which also is limited by tumor identification and clear tumor margin visualization. Mucin4 (MUC4) is highly expressed in pancreatic ductal adenocarcinoma (PDAC) while showing negligible expression in the normal pancreas, which makes the MUC4 antigen a promising target for NIR antibody labeling. In the present study, we used MUC4 antibodies (anti-MUC4) labeled with IRDye800CW to brightly target human pancreatic cancer liver metastases and peritoneal carcinomatosis in nude mouse models. The results demonstrated the ability of anti-MUC4-IRDye800CW to target and brightly label metastatic tumors in two human pancreatic cancer cell line mouse models. These findings could lead to a more accurate intraoperative staging of patients with PDAC and help to determine which patients should undergo resection.

## 1. Introduction

Pancreatic cancer is a highly lethal disease in which mortality closely parallels incidence. Pancreatic cancer is often asymptomatic in most patients until it progresses to a locally advanced stage or spreads to distant organs [1]. The most common type of pancreatic cancer is pancreatic ductal adenocarcinoma (PDAC), which accounts for 90% of cases [2]. It also accounts for more than 85% of all malignant pancreatic exocrine tumors [3].

In the advanced stages of pancreatic cancer, cancer cells disseminate from a primary lesion to distal organs, which is the major reason for cancer lethality. A variety of cellular mechanisms are involved in the dissemination of cancer cells. These include colluding or invading with stroma, modulating the tumor microenvironment, and evolving resistance to therapeutics which can enhance metastasis [4,5,6,7].

Almost 50% of pancreatic cancer patients have distant metastasis on initial presentation, with the liver being the most common site [8,9]. Prognosis is poor for pancreatic cancer patients with liver metastasis, with a 5-year survival rate of less than 1% and only 5 months of median overall survival [10]. The second most common site for PDAC metastasis is the peritoneum. PDAC can metastasize within the peritoneal cavity, driven by the tumor microenvironment [11]. Nine percent of PDAC patients present with peritoneal carcinomatosis at diagnosis [12]. In autopsies of patients who died following potentially curative resection, one-third had peritoneal dissemination [13]. A total of 50% of patients who died from PDAC, with or without surgery, or systemic treatment, had peritoneal metastasis [14].

Accurate staging of pancreatic cancer is important to determine which patients are candidates for pancreatectomy. Patients will often undergo a diagnostic laparoscopy before a resection of PDAC, and if liver or peritoneal metastases are found, then surgery of the primary tumor is aborted. With fluorescence labeling of pancreatic metastases, we have previously shown that the accuracy of diagnostic laparoscopy can be improved in mouse models [15,16].

Mucins are multifunctional glycoproteins that coat epithelial cell surfaces within the gastrointestinal tract [17]. Additionally, mucins are involved in the pathogenesis of pancreatic cancer [18]. Mucins are divided into two categories: secreted and membrane-bound. Mucin 4 (MUC4) is membrane-bound, has a high expression in pancreatic cancer, and is involved in pathobiology and aggressiveness [19]. Pancreatic ducts do not express MUC4 at the initial stage of pancreatic intraepithelial neoplasia. MUC4 expression steadily rises as the disease progresses [20]. MUC4 expression is involved in the neoplastic transformation and progression of pancreatic cancer; it influences the interactions and communications between the tumor microenvironment and cancer cells [21].

MUC4 antibodies conjugated with a fluorescent dye were effective in labeling primary pancreatic cancer in orthotopic nude mouse models [22]. In the present study, we aimed to brightly label pancreatic cancer liver metastasis and peritoneal carcinomatosis in nude mouse models. The present study shows that fluorescent MUC4 antibodies brightly label metastatic pancreatic cancer in nude mouse models.

## 2. Materials and Methods

### 2.1. Mice

Male and female nude mice aged 6 to 8 weeks were purchased from The Jackson Laboratory (Bar Harbor, ME, USA) and used in equal numbers in the present study. A cocktail in the range of 200–300 µL containing ketamine, xylazine, and phosphate-buffered saline (PBS) was injected via intraperitoneal injection (IP) before any surgical procedures. Mice were provided buprenorphine diluted in PBS (dosage: 0.05 mg/kg) via subcutaneous injection for postoperative pain control after surgery. At the end of the experiment, mice were sacrificed by CO_2_ inhalation, and then mouse death was confirmed by cervical dislocation. All animal studies were approved by the San Diego Veterans Administration Medical Center Institutional Animal Care and Use Committee (IACUC) protocol A17-020, and the University of California San Diego IACUC protocol S99001.

### 2.2. Cell Culture

SW1990 and CD18/HPAF cell lines are well-established pancreatic cancer cell lines derived from a human pancreatic ductal adenocarcinoma of a 56-year-old male [23] and 44-year-old male, respectively [24]. Cell lines were maintained in Dulbecco’s Modified Eagle’s Medium (DMEM) high-glucose media, as previously reported [22]. A total of 10% FBS and 1% penicillin–streptomycin was added to the DMEM. Cells were grown at 37 °C and 5% CO_2_.

### 2.3. Xenograft Establishment of Pancreatic Cancer Cell Lines

Mice were injected subcutaneously with SW1990 or CD18/HPAF cells (1 × 10^6^) in the contralateral flanks. When tumors of approximately 1 cm^3^ size developed, they were harvested, divided into approximately 1 mm^3^ fragments, and implanted into additional mice subcutaneously in two or four flanks. Subsequent tumors were used as stock for orthotopic implantation, as described below.

### 2.4. Liver Orthotopic Implant Metastasis Model Establishment

Surgical orthotopic implantation (SOI) was used to establish pancreatic tumors in the liver of nude mice as a liver metastasis mouse model, as previously described [25]. Briefly, a 70% ethanol solution was sprayed over the ventral surface of anesthetized mice. An incision was made on the midline to exteriorize the liver. A small flap was raised on the surface of the liver with scissors. A fragment of SW990 or CD18/HPAF tumors, approximately 0.5 mm^3^, was inserted inside the flap. The liver was re-inserted in the abdomen. The incision was closed with 6-0 vicryl sutures. Buprenorphine (150 µL) was injected subcutaneously to control postoperative pain.

### 2.5. Peritoneal Carcinomatosis Mouse Model Establishment

After nude mice were first anesthetized, SW1990 or CD18/HPAF (1 × 10^6^) cells were injected into the peritoneal cavity using a 28-gauge needle syringe.

### 2.6. Antibody-Conjugate Preparation and Administration

MUC4 antibody and IgG control were conjugated with IRDye800CW as described in our previous publication [22]. The IRDye800CW protein labeling kit (LI-COR Biosciences, Lincoln, NE, USA) was used to conjugate IRDye800CW to antibodies. To adjust the pH to 8.5, 100 µL of potassium phosphate buffer (pH 9) was added to 1 mg of anti-MUC4 antibody (1 mg/mL) or IgG. IRDye800CW (0.1 mg) was dissolved in 50 µL of nanopure water, and 12 µL of IRDye800CW solution was mixed with the antibody solution. The mixture was incubated in a dark at room temperature for 2 h (Appendix A). Unbound IRDye800CW was removed using a Zeba desalting column (ThermoFisher Scientific, Waltham, MA, USA). The dye-to-protein (D/P) ratio and protein concentration were determined using an Evolution 220 absorbance spectrophotometer (Thermo Fisher Scientific, Waltham, MA, USA). Tumors were palpable 4 weeks after liver implantation and peritoneal cell injection. A total of 50 µg anti-MUC4-IR800 or 50 µg IgG-IR800 control was administered via tail vein injection into metastatic nude mouse models of SW1990 and CD18/HPAF. Mice were sacrificed after 72 h, and a midline laparotomy was performed for bright-light and near-infrared (NIR) imaging.

### 2.7. Near-Infrared Imaging and Fluorescence Image Analysis

For NIR imaging, the Pearl Trilogy Small Animal Imaging System (LI-COR Biosciences, Lincoln, NE, USA) was used. Bright-light images were obtained using the white-light channel. Region of interests (ROIs) were drawn around the tumor and normal liver in the liver implant mouse model with the system’s software, while observing the images from the bright-light channel and NIR channel simultaneously. ROIs were drawn around normal tissues while looking at the bright-light channel, and ROIs for tumors were drawn based on bright NIR images. Boundaries were determined by the high NIR signal around the tumors (Appendix A). For all images of both the models, treated with either anti-MUC4-IR800 or IgG-IR800, the same brightness and contrast settings were used. For the peritoneal carcinomatosis mouse model, ROIs were drawn around every tumor present in the abdomen and the normal liver was used for background signal analysis. The Pearl system was used to calculate the mean fluorescence intensity (mFI) of the NIR signal of the ROIs drawn around each tumor and normal liver. Tumor-to-background (TBR) ratios were calculated by dividing the mFI of the tumors by the normal liver for the liver metastasis mouse model. For the peritoneal carcinomatosis mouse model, the mFI of each tumor was added and divided by the mFI of the normal liver for the calculation of the tumor-to-liver ratio (TLR).

### 2.8. Biodistribution

After NIR and bright-light images were obtained, a necropsy was performed on each mouse to collect the organs to analyze antibody accumulation in them. Collected tissues included the following: tumor, liver, pancreas, spleen, stomach, cecum, kidney, lung, and ear. The collected tissues were imaged using both the NIR channel and bright-light channel of the Pearl Triology instrument to determine the presence of fluorescent antibodies as described above.

### 2.9. Immunohistochemistry

Collected tumors were fixed in formalin for at least 72 h, embedded in paraffin cubes, and sectioned. Standard protocols were followed for hematoxylin and eosin (H & E) staining. MUC4 and IgG immunohistochemistry were carried out as previously described [22]. Serial sections were incubated overnight at 4 °C with either an anti-MUC4 antibody (2 µg/mL) or an IgG isotype control, followed by addition of a peroxidase-labeled secondary antibody with signal detection using the DAB substrate (Vector Universal Staining Kit, VectorLab, Newark, CA, USA).

### 2.10. Statistical Analysis

For the experimental groups (anti-MUC4-IR800), a sample size in the range of 4–5 mice per metastasis model was chosen. For the control group (IgG-IR800), a sample size of 3 mice per metastasis model was used. Power calculations were performed to determine the minimum number of subjects for adequate study. Means from the experimental and control groups are presented with standard deviations. Using an alpha value of 0.05, a power of 90% was achieved. A *p*-value of <0.05 was used as a predetermined cutoff for statistical significance.

## 3. Results

### 3.1. Targeting of Pancreatic Cancer Metastasis in a Liver Implant Model

NIR images were correlated with bright-light images, which brightly targeted SW1990 and CD18/HPAF tumors by anti-MUC4-IR800 in liver-implant nude mouse models (Figure 1A,D). The average mFI was 0.500 (±0.070) for SW1990 and 0.625 (±0.175) for CD18/HPAF tumors labeled with anti-MUC4-IR800. The average mean TBRs were 2.273 (±0.605) and 2.418 (±0.693) for SW1990 and CD18/HPAF, respectively, in mice injected with anti-MUC4-IR800 (Figure 1C,F and Table 1). SW1990 and CD18/HPAF liver-implanted mice injected with the control antibody IgG-IR800 showed a low mFI of 0.183 (±0.078) and 0.188 (±0.083) for SW1990 and CD18/HPAF tumors, respectively (Figure 1B,E). The TBRs for IgG-IR800 were 1.319 (±0.309) and 1.423 (±0.465), for SW1990 and CD18/HPAF, respectively, resulting from similar mFIs in tumors and normal liver, which show no specificity for tumors (Figure 1C,F).

### 3.2. Targeting Pancreatic Cancer Metastasis in a Peritoneal Carcinomatosis Mouse Model

In peritoneal carcinomatosis nude mouse models of SW1990 and CD18/HPAF, all tumors were brightly targeted by anti-MUC4-IR800. The average total mFI for peritoneal tumors was 1.65 (±0.545) and 2.068 (0.268) for SW1990 and CD18/HPAF tumors, respectively (Figure 2A,D, Table 2). Average TLRs were 9.048 (±3.383) and 7.736 (±2.891) for SW1990 and CD18/HPAF tumors, respectively (Figure 2C,F, Table 2). The peritoneal tumors of mice injected with IgG-IR800 had an mFI of 0.0879 (±0.0133) and 0.485 (±0.216) for SW1990 and CD18/HPAF tumors, respectively (Figure 2B,E). TLRs were 1.051 (±0.566) and 1.960 (±0.488) for SW1990 and CD18/HPAF, respectively, from the non-specific binding of IgG-IR800 (Figure 2C,F).

### 3.3. Flourescence Biodistribution of Anti-MUC4-IR800 and IgG-IR800

Biodistribution studies were performed to analyze the accumulation of fluorescent antibodies (anti-MUC4-IR800 or IgG-IR800) in different organs. The highest fluorescence signal for anti-MUC4-IR800 accumulation was observed for SW1990 and CD18/HPAF tumors in liver implant metastases and peritoneal carcinomatosis (Figure 3). The second highest signals were present in normal liver and kidney due to lack of clearance. Other organs showed relatively very low signals. Negligible NIR signals seen in other organs may have be due to the non-specific binding or inflammation. SW1990 and CD18/HPAF models injected with the IgG-IR800 control antibody showed similar low signals in the liver and tumors confirming its non-selectivity.

The average mFI and standard deviation in tumors and other organs in the liver metastasis mouse model of SW1990 injected with anti-MUC4-IR800 were as follows: tumor: 0.415 (±0.042), liver: 0.227 (±0.022), ear: 0.072 (±0.019), kidney: 0.140 (±0.011): lung: 0.087 (±0.014), stomach: 0.105 (±0.019), spleen: 0.082 (±0.018), cecum: 0.057 (±0.010), and pancreas: 0.092 (±0.014). The average mFI and standard deviation in tumors and other organs in the injected control mice with IgG-IR800 were as follows: tumor: 0.110 (±0.058), liver: 0.105 (±0.001), ear: 0.022 (±0.008), kidney: 0.038 (±0.015): lung: 0.033 (±0.006), stomach: 0.025 (±0.011), spleen: 0.019 (±0.007), cecum: 0.027 (±0.03), and pancreas: 0.022 (±0.001).

The average mFI and standard deviation in tumors and other organs in the liver metastasis mouse model of CD18/HPAF injected with anti-MUC4-IR800 were as follows: tumor: 0.443 (±0.129), liver: 0.293 (±0.078), ear: 0.085 (±0.014), kidney: 0.136 (±0.026): lung: 0.123 (±0.034), stomach: 0.083 (±0.011), spleen: 0.114 (±0.013), cecum: 0.075 (±0.023), and pancreas: 0.119 (±0.024). The average mFI and standard deviation in tumors and other organs injected in control mice with IgG-IR800 were as follows: tumor: 0.166 (±0.087), liver: 0.105 (±0.001), ear: 0.022 (±0.008), kidney: 0.038 (±0.015): lung: 0.033 (±0.006), stomach: 0.025 (±0.011), spleen: 0.019 (±0.007), cecum: 0.027 (±0.03), and pancreas: 0.022 (±0.001).

The average mFI and standard deviation in tumors and other organs in the peritoneal carcinomatosis mouse model of SW1990 injected with anti-MUC4-IR800 were as follows: tumor: 1.653 (±0.545), liver: 0.232 (±0.079), ear: 0.071 (±0.034), kidney: 0.130 (±0.047): lung: 0.102 (±0.056), stomach: 0.119 (±0.071), spleen: 0.101 (±0.046), cecum: 0.073 (±0.033), and pancreas: 0.094 (±0.025). The average mFI and standard deviation in tumors and other organs injected in the control mice with IgG-IR800 were as follows: tumor: 0.110 (±0.034), liver: 0.105 (±0.015), ear: 0.019 (±0.006), kidney: 0.038 (±0.017): lung: 0.028 (±0.013), stomach: 0.023 (±0.001), spleen: 0.019 (±0.001), cecum: 0.028 (±0.017), and pancreas: 0.032 (±0.006).

The average mFI and standard deviation in tumors and other organs in the peritoneal carcinomatosis mouse model of CD18/HPAF injected with anti-MUC4-IR800 were as follows: tumor: 2.06 (±0.268), liver: 0.378 (±0.184), ear: 0.058 (±0.035), kidney: 0.135 (±0.026): lung: 0.122 (±0.044), stomach: 0.181 (±0.0231), spleen: 0.150 (±0.056), cecum: 0.122 (±0.053), and pancreas: 0.081 (±0.028). The average mFI and standard deviation in tumors and other organs injected in the control mice with IgG-IR800 were as follows: tumor: 0.485 (±0.216), liver: 0.236 (±0.058), ear: 0.007 (±0.001), kidney: 0.038 (±0.015): lung: 0.022 (±0.003), stomach: 0.035(±0.021), spleen: 0.028 (±0.001), cecum: 0.040 (±0.011), and pancreas: 0.033 (±0.001).

### 3.4. Confirmation of MUC4 Expression in Tumors by Immunohistochemistry

Immunohistochemical staining confirmed the presence of MUC4 expression in all the tumors collected from liver implant (SW1990 and CD18/HPAF) and peritoneal carcinomatosis (SW1990 and CD18/HPAF) models. The high expression of MUC4 is indicated by strong brown staining in Figure 4 in both tumor types in both mouse models. H & E staining confirmed the histology in both tumor models (Figure 4).

## 4. Discussion

In the present study, we tested two major types of pancreatic cancer metastasis, liver metastasis and peritoneal carcinomatosis, for selective tumor targeting with anti-MUC4-IR800. The high selectivity of anti-MUC4-IR800 for the SW1990 and CD18/HPAF tumors produced high tumor-to-background ratios, 2.27 and 2.41, respectively, which clearly distinguished tumors from normal liver. In the peritoneal carcinomatosis models, SW1990 and CD18/HPAF tumors were fluorescently well-distinguished from the other organs, including the liver. Tumors as small as 1 mm were fluorescently distinguished in the peritoneal cavity by targeting with anti-MUC4-IR800 in SW1990 and CD18/HPAF peritoneal carcinomatosis nude mouse models. Histology results confirmed pancreatic cancer in the NIR fluorescent tumors. Normal tissue was confirmed in non-NIR fluorescent labeled tissues confirming the specificity of anti-MUC4-IR800 for targeting pancreatic metastasis. Strong MUC4 staining in collected tumors and negligible MUC4 staining in normal organs further strengthen the validity of using anti-MUC4-IR800 for tumor labeling.

Fluorescence-guided surgery (FGS) using tumor-targeting near-infrared probes has emerged as a promising technique for intraoperative tumor detection and resection [26,27,28]. IRDye800CW is a non-toxic IR dye that has been used as a fluorophore for conjugation with tumor specific antibodies for the detection of tumors [29,30,31,32,33]. Clinical trials have been ongoing on for tumor targeting probes, including CEA [34]. However, high fluorescence accumulation in the liver and bladder can result which can interfere when visualizing primary or metastatic tumors. In the present study, by using anti-MUC4-IR800, we were able to achieve an almost 2.5-times-lower liver background signal compared to the liver metastatic tumor despite imaging at an early timepoint. High tumor-to-liver fluorescence ratios resulted from the high specificity of anti-MUC4-IR800 for the tumor. Bright targeting with a low background signal can help surgeons to identify a tumor and its resection margins.

The limitations of this study include using only one imaging modality to visualize metastatic tumors. Although the pathology results in the present study confirm the tumor and normal tissue characteristics, the addition of bioluminescence imaging with luciferase-labeled cancer cells can allow for the co-localization of tumors [25]. In future studies, SW1990 and CD18/HPAF human pancreatic cancer cell lines can be modified with luciferase or various color fluorescent proteins, which can further confirm the specific targeting of anti-MUC4-IR800. Secondly, FGS using anti-MUC4-IR800 may not be effective for the very early stages of metastasis. However, tumor-targeting probes such as those used in the present study can still increase the detection rate of small metastatic lesions that may be missed by the naked eye or by preoperative imaging. Future studies will also be conducted to determine the depth of penetration of MUC4-IR800.

While mouse models offer valuable insights into targeting tumors, important limitations include differences in immune system function, the tumor microenvironment, drug metabolism, and genetic heterogeneity between mice and humans. Preclinical efficacy observed in murine models does not always predict clinical outcomes due to these interspecies differences. Addressing these challenges will ultimately require careful validation in well-designed clinical trials.

Surgical resection is an important aspect of treatment for pancreatic cancer, which requires the clear identification of all tumors from normal tissues. The present study shows that anti-MUC4-IR800 can identify difficult-to-image liver metastasis as well as a range of different-sized peritoneal tumors. These findings could lead to more accurate intraoperative staging of patients with PDAC and help to determine which patients should undergo resection.

## 5. Conclusions

The present study demonstrates that anti-MUC4-IR800 can detect and distinguish human pancreatic cancer metastases in two mouse models: liver metastasis and peritoneal carcinomatosis. The present study suggests anti-MUC4-IR800 labeling can be used to better identify liver metastasis and peritoneal metastatic pancreatic cancers from surrounding normal tissues and improve operative staging for patients with pancreatic cancer.

## Figures and Tables

**Figure 1 cancers-17-02031-f001:**
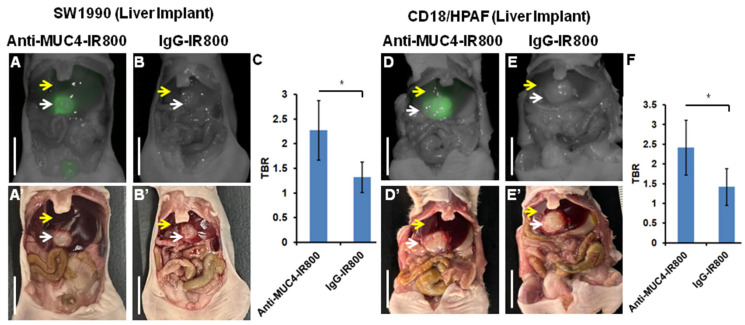
Labeling by anti-MUC4-IR800 and IgG-IR800 of liver metastasis models of SW1990 (left panel) and CD18/HPAF pancreatic cancer (right panel). (**A**) NIR and (**A’**) bright-light imaging of SW1990 liver metastasis models labeled with 50 µg anti-MUC4-IR800. (**B**) Non-specific NIR labeling with 50 µg IgG-IR800 and (**B’**) bright-light imaging. (**C**) Average tumor-to-background ratio (TBR) of mice treated with 50 µg anti-MUC4-IR800, n = 5, and mice treated with 50 µg IgG-IR800, n = 3. * *p*-value = 0.04 (**D**) NIR and (**D’**) bright-light imaging of CD18/HPAF liver metastasis models labeled with 50 µg MUC4-IR800. (**E**) Non-specific NIR labeling with 50 µg IgG-IR800 and (**E’**) bright-light imaging. (**F**) The average tumor-to-background ratio (TBR) of mice treated with 50 µg MUC4-IR800, n = 5, and those treated with 50 µg IgG-IR800, n = 3. * *p*-value = 0.03. Yellow arrow: normal liver and white arrows: tumors. Scale bar: 1 cm.

**Figure 2 cancers-17-02031-f002:**
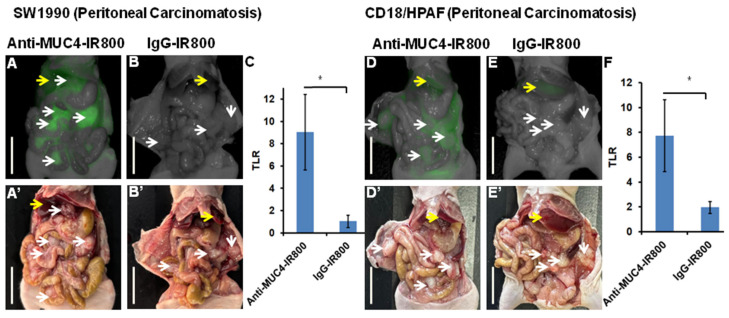
Labeling of pancreatic cancer nude mouse model of peritoneal carcinomatosis of SW1990 (left panel) and CD18/HPAF (right panel) labeled with anti-MUC4-IR800 or IgG-IR800. (**A**) NIR and (**A’**) bright-light imaging of SW1990 peritoneal carcinomatosis metastasis models labeled with 50 µg anti-MUC4-IR800. (**B**) Non-specific NIR labeling with 50 µg IgG-IR800 and (**B’**) bright-light imaging. (**C**) Average tumor-to-liver ratio (TLR) of mice treated with 50 µg anti-MUC4-IR800, n = 4 and mice treated with 50 µg IgG-IR800, n = 3, * *p*-value = 0.02. (**D**) NIR and (**D’**) bright-light imaging of CD18/HPAF peritoneal carcinomatosis metastasis models labeled with 50 µg MUC4-IR800. (**E**) Non-specific NIR labeling with 50 µg IgG-IR800 and (**E’**) bright-light imaging. (**F**) Average tumor-to-liver ratio (TLR) of mice treated with 50 µg MUC4-IR800, n = 4, and mice treated with 50 µg IgG-IR800, n = 3, * *p*-value = 0.02. Yellow arrow: normal liver and white arrows: tumors. Scale bar: 1 cm.

**Figure 3 cancers-17-02031-f003:**
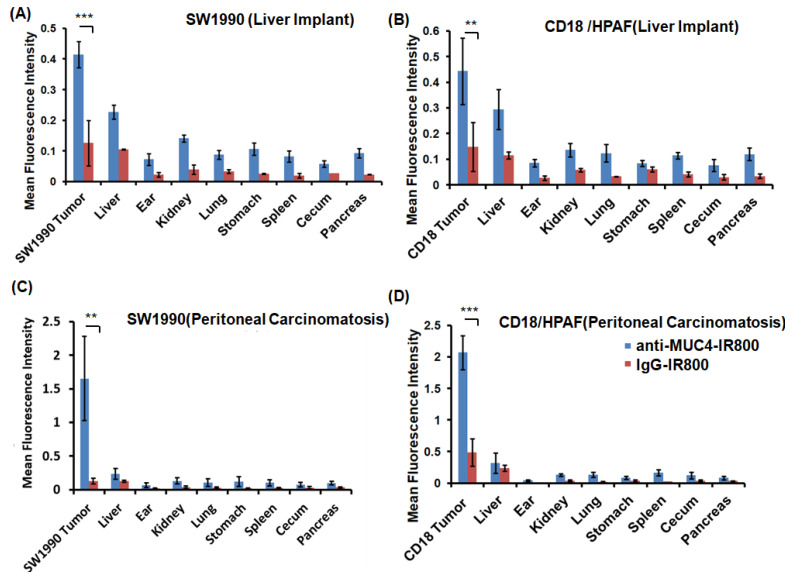
Fluorescence biodistribution of anti-MUC4-IR800 in (**A**,**B**) in liver implant metastases in SW1990 and CD18/HPAF models, respectively, indicated by blue bars. Fluorescence biodistribution of IgG-IR800 in (**A**,**B**) in liver implant metastasis model in SW1990 and CD18/HPAF models, respectively, indicated by red bars. Fluorescence biodistribution of anti-MUC4-IR800 in (**C**,**D**) in peritoneal carcinomatosis in SW1990 and CD18/HPAF, respectively, indicated by blue bars. Fluorescence biodistribution of IgG-IR800 in (**C**,**D**) in peritoneal carcinomatosis in SW1990 and CD18/HPAF models, respectively, indicated by red bars. *p* values are 0.0007 (**A**), 0.012 (**B**), 0.007 (**C**), and 0.0003 (**D**). ** and *** indicates *p*-value significance level.

**Figure 4 cancers-17-02031-f004:**
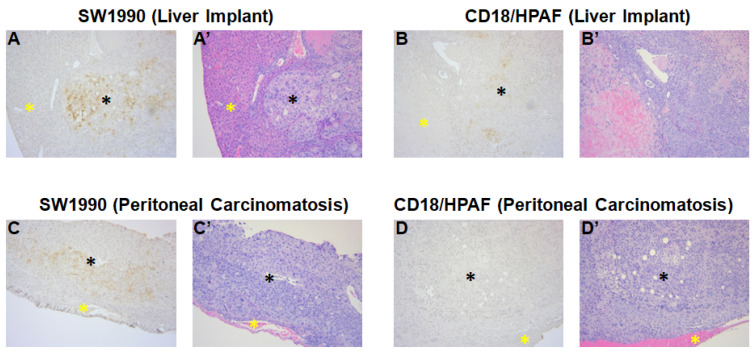
Immunohistochemical staining images (10×) of MUC4 in liver implant and peritoneal carcinomatosis nude mouse models. (**A**,**A’**) indicates MUC4-staining of SW1990 cells and H & E staining for the same tumor from the liver implant mouse model. Pancreatic tumor is denoted by a black asterisk and normal liver is denoted by a yellow asterisk. (**B**,**B’**) indicates the MUC4-stained CD18/HPAF and H & E staining for the same tumor from the liver implant mouse model. Pancreatic tumor is denoted by a black asterisk and normal liver is denoted by a yellow asterisk. (**C**,**C’**) indicates MUC4-staining of SW1990 cells and H & E staining for the same tumor from the peritoneal carcinomatosis mouse model. Pancreatic tumor is denoted by a black asterisk and normal peritoneal tissue is denoted by a yellow asterisk. (**D**,**D’**) indicates MUC4-staining of CD18/HPAF cells and H & E staining for the same tumor from the peritoneal carcinomatosis mouse model. Pancreatic tumor is denoted by a black asterisk and normal peritoneal tissue is denoted by a yellow asterisk.

**Table 1 cancers-17-02031-t001:** Mean fluorescence intensity values of the liver-implanted tumors and normal liver for individual mice bearing SW1990 or CD18/HPAF tumors injected with anti-MUC4-IR800. Calculated tumor-to-background ratios; SD: standard deviation.

Mouse	SW1990 Tumor (mFI)	Liver (mFI)	SW1990 Tumor/Liver (TBR)	CD18/HPAF Tumor (mFI)	Liver (mFI)	CD18/HPAF Tumor/Liver (TBR)
1	0.601	0.198	3.035	0.359	0.171	2.099
2	0.5	0.214	2.336	0.878	0.318	2.761
3	0.447	0.207	2.159	0.734	0.203	3.615
4	0.455	0.291	1.563	0.611	0.325	1.880
5	0.357	0.326	1.095	0.544	0.313	1.738
Average (±SD)	0.50075 (±0.070)	0.2275 (±0.042)	2.273 (±0.605)	0.6252 (±0.175)	0.266 (±0.065)	2.418 (±0.693)
*p*-value	0.0007		0.04	0.012		0.03

**Table 2 cancers-17-02031-t002:** Total mean fluorescence intensity values of the peritoneal carcinomatosis tumors and normal liver for individual mice bearing SW1990 and CD18/HPAF tumors and injected with anti-MUC4-IR800. Calculated tumor-to-liver ratios. SD: Standard deviation.

Mouse	SW1990 Tumor (mFI) (Total)	Liver (mFI)	SW1990 Tumor/Liver (TLR)	CD18/HPAF Tumor (mFI) (Total)	Liver (mFI)	CD18/HPAF Tumor/Liver (TLR)
1	2.404	0.171	14.058	1.959	0.16	12.243
2	1.752	0.181	9.679	2.454	0.587	4.180
3	0.871	0.113	7.707	2.144	0.300	7.146
4	1.586	0.334	4.748	1.718	0.233	7.373
Average (±SD)	1.653 (±0.545)	0.334 (0.0817)	9.048 (±3.383)	2.068 (± 0.268)	0.320 (± 0.161)	7.736 (±2.891)
*p*-value	0.007		0.02	0.0003		0.02

## Data Availability

The original contributions presented in this study are included in the article/Appendix A; further inquiries can be directed to the corresponding author.

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
