# Peer review of "Mucin4 (MUC4) Antibody Labeled with an NIR Dye Brightly Targets Pancreatic Cancer Liver Metastases and Peritoneal Carcinomatosis"

_cancers, 2025, doi:10.3390/cancers17122031_

Round 1

Reviewer 1 Report

Comments and Suggestions for Authors

The manuscript "Mucin4 (MUC4) Antibody Labeled with an NIR Dye Brightly Targets Pancreatic Cancer Liver Metastases and Peritoneal  Carcinomatosis" has scientific soundness and has academic and clinical importance. Despite the aims of the manuscript are not clear, the idea seems to be: "to target pancreatic cancer liver metastases and peritoneal carcinomatosis in nude mouse models", there are also some concerns regarding the manuscript: 

  • The abstract does not have materials and methods, results and discussion/conclusion. Please rewrite it.
  • The introduction does not provide any information regarding the antobody labeled with the NIR Dye Brightly. 
  • The introduction do not have any aims
  • The materials and methods do not provide sufficient information regarding the mice , the cell culture,  and the antibodies used for the immunohistochemistry. The manuscript should be reproducible and in these conditions is not. 
  • The discussion must be rewritten. The authors do not discuss any of the results, do not explain the importance of the labeled NIR Dye Brightly in targeting pancreatic cancer liver metastases. 
  • The authors fail to explain why this methodology  can improve operative staging as claimed in the conclusions
  • The references are insufficient. Please update it. 

Author Response

The manuscript "Mucin4 (MUC4) Antibody Labeled with an NIR Dye Brightly Targets Pancreatic Cancer Liver Metastases and Peritoneal  Carcinomatosis" has scientific soundness and has academic and clinical importance. Despite the aims of the manuscript are not clear, the idea seems to be: "to target pancreatic cancer liver metastases and peritoneal carcinomatosis in nude mouse models", there are also some concerns regarding the manuscript: 

        Comment 1: The abstract does not have materials and methods, results and discussion/conclusion. Please rewrite it.

Response 1: Thank you for the suggestion. Revised abstract is now included in revised manuscript.

      Comment 2: The introduction does not provide any information regarding the antibody labeled with the NIR Dye Brightly. 

Response 2: Included in revised introduction.

       Comment 3: The introduction do not have any aims

Response 3: Included in revised manuscript.  

      Comment 4: The materials and methods do not provide sufficient information regarding the mice , the cell culture,  and the antibodies used for the immunohistochemistry. The manuscript should be reproducible and in these conditions is not. 

Response 4:  Included in revised manuscript.

Comment 5: The discussion must be rewritten. The authors do not discuss any of the results, do not explain the importance of the labeled NIR Dye Brightly in targeting pancreatic cancer liver metastases. 

Response 6: Included in revised manuscript, discussion section, page 10.

 Comment 7: The authors fail to explain why this methodology  can improve operative staging as claimed in the conclusions

Response 7: Included in revised manuscript, discussion section, page 10.

Comment 8: The references are insufficient. Please update it. 

Response 8: More references are now included in revised manuscript.

Reviewer 2 Report

Comments and Suggestions for Authors

This article presents the first validation of the targeting ability of MUC4-IR800 in liver metastasis and peritoneal metastasis models, filling a research gap in the detection of pancreatic cancer metastasis by fluorescently labeled antibodies. Suggests the possibility of improving pancreatic cancer staging by intraoperative fluorescence imaging, which meets the unmet clinical needs.

Where the article needs improvement:

1.the small sample size of some experimental groups (e.g., n=3) may affect the statistical validity, which needs to be supplemented with the basis of sample size calculation.

2.other fluorescent labeling antibodies (e.g., anti-CEA or anti-EGFR) in pancreatic cancer should be cited and compared to highlight the advantages of this study.

3.The arrows in Figures 1-4 (yellow/white arrows) should be clearly explained in the figure notes (e.g., “yellow arrow: normal liver tissue; white arrow: tumor”).

4.The studies on MUC4 in metastatic pancreatic cancer in the past 3 years should be added.

5.Sensitivity analysis of fluorescence imaging should be added: e.g. minimum detectable tumor size, penetration depth data.

6.Discuss the risk of false positives/false negatives: e.g. whether normal tissue (e.g. sites of inflammation) may bind MUC4-IR800 non-specifically.

Author Response

This article presents the first validation of the targeting ability of MUC4-IR800 in liver metastasis and peritoneal metastasis models, filling a research gap in the detection of pancreatic cancer metastasis by fluorescently labeled antibodies. Suggests the possibility of improving pancreatic cancer staging by intraoperative fluorescence imaging, which meets the unmet clinical needs.

Where the article needs improvement:

Comment 1: The small sample size of some experimental groups (e.g., n=3) may affect the statistical validity, which needs to be supplemented with the basis of sample size calculation.

Response 1:  Included in revised manuscript.

Comment 2:  Other fluorescent labeling antibodies (e.g., anti-CEA or anti-EGFR) in pancreatic cancer should be cited and compared to highlight the advantages of this study.

Response 2:  Included in revised manuscript, discussion section, page 10.

Comment 3: The arrows in Figures 1-4 (yellow/white arrows) should be clearly explained in the figure notes (e.g., “yellow arrow: normal liver tissue; white arrow: tumor”).

Response 3: Included in revised manuscript.

Comment 4:  The studies on MUC4 in metastatic pancreatic cancer in the past 3 years should be added.

Response 4: Studies on MUC4 in metastatic pancreatic cancer have been added.

Comment 5: Sensitivity analysis of fluorescence imaging should be added: e.g. minimum detectable tumor size, penetration depth data.

Response 5:  The minimum detectable tumor size has now been discussed in manuscript. We have discussed future studies to determine the penetration depth of MUC4-IR800.

Comment 6:  Discuss the risk of false positives/false negatives: e.g. whether normal tissue (e.g. sites of inflammation) may bind MUC4-IR800 non-specifically.

Response 6: Discussed in revised manuscript.

Reviewer 3 Report

Comments and Suggestions for Authors

## Suggestions for Authors

### Scientific Content
1. The study demonstrates promising results for MUC4 antibody targeting of pancreatic cancer metastases, but would benefit from addressing several points:
   - Consider including a discussion about potential translation barriers from mouse models to human applications
   - Address potential limitations regarding antibody specificity and background signal in clinical settings
   - Compare your tumor-to-background ratios with other fluorescence-guided surgery approaches in the literature

2. Methods and Results:
   - Include power calculations to justify your sample sizes (n=3-5 per group)
   - Consider adding more detailed biodistribution analysis, including quantification of uptake in other organs
   - Provide clearer explanation of how tumor boundaries were determined when calculating regions of interest
   - Include more discussion on the detection threshold (minimum tumor size detectable)

3. Discussion and Conclusions:
   - Expand on potential clinical applications beyond diagnostic laparoscopy
   - Address how this approach compares to current clinical staging methods
   - Discuss potential limitations such as penetration depth of NIR fluorescence in human tissues
   - Consider including a paragraph on future directions and ongoing/planned clinical studies

### Manuscript Structure and Presentation
1. Figures and Tables:
   - Figure 1 and 2 should include scale bars on the images
   - Consider adding a supplementary figure showing the antibody conjugation process
   - Table 1 and 2 could be enhanced with additional statistical information (p-values for all comparisons)

2. References:
   - Update references to include more recent work (several are from 2016-2017)
   - Include references to clinical trials using NIR-guided surgery for pancreatic cancer

Comments on the Quality of English Language

The manuscript requires moderate language editing. There are several grammatical errors and awkward phrasings throughout. Some specific examples:
- Line 31-32: "Most patients with pancreatic cancer remain asymptomatic until the disease becomes locally advanced or metastasizes to other organs" (awkward construction)
- Lines 45-46: "PDAC metastasizes quickly in the peritoneal cavity, contributing to poor prognosis. It is mainly driven by tumor cell-intrinsic plasticity" (unclear pronoun reference)
- Several instances of passive voice could be converted to active voice

Author Response

  1. The study demonstrates promising results for MUC4 antibody targeting of pancreatic cancer metastases, but would benefit from addressing several points:
    Comment 1: Consider including a discussion about potential translation barriers from mouse models to human applications

Response 1: Included in revised discussion section on page 10.

Comment 2: Address potential limitations regarding antibody specificity and background signal in clinical settings

Response 2: Included in revised discussion section, page 10.

Comment 3: Compare your tumor-to-background ratios with other fluorescence-guided surgery approaches in the literature

Response 3: Included in revised manuscript, discussion section, page 10.

Methods and Results:
Comment 4: Include power calculations to justify your sample sizes (n=3-5 per group)

Response 4: A statistical section has been added to the revised manuscript.

Comment 5:  Consider adding more detailed biodistribution analysis, including quantification of uptake in other organs

Response: Included in revised manuscript, results section, page 7.
Comment 6: Provide clearer explanation of how tumor boundaries were determined when calculating regions of interest

Response: Included in revised manuscript, material and methods section, page 4 and figure S2.

Comment 7: Include more discussion on the detection threshold (minimum tumor size detectable)

Response 7: Included in revised manuscript, discussion section, page 9

  1. Discussion and Conclusions:
    Comment 8: Expand on potential clinical applications beyond diagnostic laparoscopy

Response 8:  We have expanded on clinical applications beyond diagnostic laparoscopy in the revised discussion.

Comment 9: Address how this approach compares to current clinical staging methods

Response 9:  This has now been added to the revised discussion.

Comment 10: Discuss potential limitations such as penetration depth of NIR fluorescence in human tissues

Response 10: The potential limitations are now discussed in the revised manuscript.

Comment 11:  Consider including a paragraph on future directions and ongoing/planned clinical studies

Response 11: Future directions and planned clinical studies are now discussed in the revised manuscript.

### Manuscript Structure and Presentation
1. Figures and Tables:
Comment 12: Figure 1 and 2 should include scale bars on the images

Response 12: Included in revised manuscript.

Comment 13: Consider adding a supplementary figure showing the antibody conjugation process

Response 13: Included in revised supplementary document, Figure S1.

Comment 14: Table 1 and 2 could be enhanced with additional statistical information (p-values for all comparisons)

Response 14: Included in revised manuscript.

  1. References:
    Comment 15: Update references to include more recent work (several are from 2016-2017)

Response 15: Update references have been added to include more recent work.

Comment 16: Include references to clinical trials using NIR-guided surgery for pancreatic cancer

Response 16: Included in revised manuscript, discussion section, page 10.

Comments on the Quality of English Language

The manuscript requires moderate language editing. There are several grammatical errors and awkward phrasings throughout. Some specific examples:

Comment 17: Line 31-32: "Most patients with pancreatic cancer remain asymptomatic until the disease becomes locally advanced or metastasizes to other organs" (awkward construction)

Response 17: Included in revised manuscript.

Comment 18: Lines 45-46: "PDAC metastasizes quickly in the peritoneal cavity, contributing to poor prognosis. It is mainly driven by tumor cell-intrinsic plasticity" (unclear pronoun reference)

Response 18: Included in revised manuscript.

Comment 19: Several instances of passive voice could be converted to active voice

Response 19: Included in revised manuscript.

Round 2

Reviewer 3 Report

Comments and Suggestions for Authors

Suggestions were operated by authors and now paper appears to be optimal for publication